# Hesperidin Anti-Osteoporosis by Regulating Estrogen Signaling Pathways

**DOI:** 10.3390/molecules28196987

**Published:** 2023-10-09

**Authors:** Hong-Yao Hu, Ze-Zhao Zhang, Xiao-Ya Jiang, Tian-Hua Duan, Wei Feng, Xin-Guo Wang

**Affiliations:** 1Jilin Medical Products Administration, Changchun 130000, China; hhyxyz@sina.com; 2School of Pharmaceutical Sciences, Quality Evaluation & Standardization Hebei Province Engineering Research Center of Traditional Chinese Medicine, Hebei University of Chinese Medicine, Shijiazhuang 050091, China; zhangzz0308@163.com (Z.-Z.Z.); jiangxiaoya0420@163.com (X.-Y.J.); duantianhua2022@163.com (T.-H.D.)

**Keywords:** hesperidin, osteoporosis, network pharmacology, molecular dynamics simulation, zebrafish

## Abstract

Osteoporosis (OP) is distinguished by a reduction in bone mass and degradation of bone micro-structure, frequently resulting in fractures. As the geriatric demographic expands, the incidence of affected individuals progressively rises, thereby exerting a significant impact on the quality of life experienced by individuals. The flavonoid compound hesperidin has been subject to investigation regarding its effects on skeletal health, albeit the precise mechanisms through which it operates remain ambiguous. This study utilized network pharmacology to predict the core targets and signaling pathways implicated in the anti-OP properties of hesperidin. Molecular docking and molecular dynamics simulations were employed to confirm the stability of the interaction between hesperidin and the core targets. The effects of hesperidin on osteoblastic cells MC3T3-E1 were assessed using MTT, ELISA, alkaline phosphatase assay, and RT-qPCR techniques. Furthermore, in vivo experiments were conducted to determine the potential protective effects of hesperidin on zebrafish bone formation and oxidative stress response. The results demonstrate that network pharmacology has identified 10 key target points, significantly enriched in the estrogen signaling pathway. Hesperidin exhibits notable promotion of MC3T3-E1 cell proliferation and significantly enhances ALP activity. ELISA measurements indicate an elevation in NO levels and a reduction in IL-6 and TNF-α. Moreover, RT-qPCR analysis consistently reveals that hesperidin significantly modulates the mRNA levels of ESR1, SRC, AKT1, and NOS3 in MC3T3-E1 cells. Hesperidin promotes osteogenesis and reduces oxidative stress in zebrafish. Additionally, we validate the stable and tight binding of hesperidin with ESR1, SRC, AKT1, and NOS3 through molecular dynamics simulations. In conclusion, our comprehensive analysis provides evidence that hesperidin may exert its effects on alleviating OP through the activation of the estrogen signaling pathway via ESR1. This activation leads to the upregulation of SRC, AKT, and eNOS, resulting in an increase in NO levels. Furthermore, hesperidin promotes osteoblast-mediated bone formation and inhibits pro-inflammatory cytokines, thereby alleviating oxidative stress associated with OP.

## 1. Introduction

Osteoporosis (OP) is a systemic skeletal disorder distinguished by alterations in bone structure and diminished bone density, leading to heightened bone fragility and vulnerability to fractures [1]. On a global scale, approximately half of women aged 50 and older, as well as one-fifth of men, face the possibility of developing OP or encountering fragility fractures [2]. The existence of OP and its related complications presents a significant burden in relation to both morbidity and mortality, thus establishing it as a matter of global public health importance. Given the rising prevalence of an aging population, projections indicate that over 320 million individuals will be at an elevated risk of fractures by the year 2040. Furthermore, it is projected that by 2050, the occurrence of hip fractures in both men and women worldwide will rise by 310% and 240%, respectively [3,4]. OP is predominantly observed in postmenopausal women, primarily due to the significant decrease in estrogen levels following menopause [5,6]. The pain and skeletal deformities resulting from OP significantly diminish the quality of life for patients and impose substantial economic burdens on both society and families.

Autophagy is a highly conserved cellular mechanism that safeguards cell homeostasis by degrading damaged or obsolete organelles, breaking down useless macromolecules or pathogens, and releasing nutrients and energy [7]. Existing evidence suggests that autophagy plays a beneficial role in the early stages of osteoblast proliferation, differentiation, and survival, while dysfunctional autophagy exacerbates mitochondrial oxidative stress and damage, leading to osteoporosis [8]. Reports have indicated that glucocorticoids inhibit autophagy in osteoblasts, which is related to the pathogenesis of glucocorticoid-induced osteoporosis [9]. Therefore, this study investigates the regulatory effects of hesperidin on dexamethasone-inhibited autophagy and cell apoptosis in osteoblasts, as well as the modulation of bone formation and oxidative stress by prednisolone in zebrafish.

Hesperidin is a bioflavonoid compound primarily found in citrus fruits such as oranges, lemons, tangerines, and grapefruits [10]. Hesperidin demonstrates a range of biological activities, encompassing anti-inflammatory, antioxidant, anti-aging, anticancer, and antibacterial properties [11,12,13,14]. An increasing body of evidence suggests that flavonoid compounds play a significant role in ameliorating skeletal diseases and promoting overall bone health, including osteoporosis, by exerting their effects through various mechanisms that enhance bone density. These mechanisms involve the inhibition of oxidative stress and inflammatory processes, thereby facilitating the formation of osteoblasts and the differentiation of osteoclasts [15,16,17]. Several studies have indicated that hesperidin possesses the ability to mitigate bone loss and promote bone defect regeneration. Significantly, the research conducted by Chiba, Horcajada, and other scholars [18,19] has yielded evidence that supports the substantial protective impact of hesperidin against bone loss in mice that underwent ovariectomy. Additionally, hesperidin exhibits therapeutic potential due to its anti-inflammatory properties. Kuo et al. [20] have discovered that hesperidin can ameliorate ligature-induced periodontitis in rats by suppressing the expression of inflammatory markers such as IL-6, IL-1β, and iNOS, as well as inhibiting the production of inflammatory mediators. Moreover, according to literature findings, hesperidin has been suggested to possess the capability to stimulate osteogenesis. In a study conducted by Miguez et al. [21], the effects of hesperidin on preosteoblast cell function, osteogenesis, and collagen matrix quality were examined. The study results revealed that hesperidin not only regulates mineralized tissue formation through the modulation of osteoblast differentiation but also controls the formation of mineralized tissue by adjusting the ratio between matrix tissue and mineral. Therefore, hesperidin may serve as a potential adjunct material for regenerative bone therapy. However, to our knowledge, there is still relatively limited research on the role of hesperidin in improving oxidative stress in the context of OP.

In this study, we employed network pharmacology methods to comprehensively analyze the mechanism of action of hesperidin in combating OP. In vitro experiments were conducted to validate the results of our network pharmacology analysis. On the one hand, we investigated the effects of hesperidin on the proliferation of osteoblast-like cells MC3T3-E1, ALP activity, and levels of inflammatory factors. On the other hand, we discovered that hesperidin could reduce the generation rate of ROS in zebrafish, alleviate oxidative stress, and promote zebrafish bone formation. Additionally, RT-qPCR was employed to demonstrate the anti-osteoporotic activity of hesperidin against core targets, which was further confirmed through molecular dynamics simulations. The following figure illustrates the schematic diagram of the research methodology employed in this study (Figure 1).

## 2. Results

### 2.1. Network Pharmacology Analysis

#### 2.1.1. Potential Target Identification of Hesperidin for OP

After merging and eliminating duplicates, a comprehensive analysis using SwissTargetPrediction and PharmMapper databases resulted in the identification of 310 target proteins associated with the compound hesperidin. Disease targets were filtered using the GeneCards, OMIM, and TTD disease databases, with “OP” and “oxidative stress” as the key search terms. By mapping the compound targets, disease targets, and corresponding genes, a Venn diagram (Figure 2A) was generated, revealing a total of 33 intersecting target proteins.

#### 2.1.2. PPI Network and Core Targets

As a result of the Venn diagram, 33 intersecting target proteins were uploaded to STRING for the purpose of building a PPI network. Data was simultaneously exported to “TSV” format and imported into Cytoscape 3.8.0 for visual analysis (Figure 2B). Using the criteria of Betweenness Centrality > 0.0205, Closeness Centrality > 0.6305, and Degree > 13, which are higher than the average values, 10 core target proteins were selected. Subsequently, a “Hesperidin-Core Targets-Pathways-Disease” network was constructed (Figure 2C).

#### 2.1.3. GO and KEGG Enrichment Analysis

With the help of the Metascape database, core targets of hesperidin against OP were enriched with GO and KEGG information. Visualization analysis was performed using bubble plots and bar charts through bioinformatics tools, as shown in Figure 2D,E. The GO biological function enrichment analysis covered three aspects: Biological Processes, Molecular Functions, and Cellular Components. Regarding Biological Processes, a comprehensive total of 217 pertinent entries were enriched, predominantly encompassing responses to estrogen, regulation of external stimulus response, and negative regulation of apoptotic signaling pathways. Molecular Functions yielded 12 relevant entries, primarily related to calcium-binding proteins, protein kinases, and transcription factor binding reactions. Concerning Cellular Components, six pertinent entries were enriched, primarily linked to the secretory granule lumen, vesicle lumen, and extracellular matrix. The KEGG pathway enrichment analysis revealed a total of 23 hesperidin anti-OP signaling pathways, potentially exerting its effects through the regulation of the Estrogen signaling pathway.

### 2.2. Molecular Docking

Using the AutoDock Vina 1.2.2 software, the binding energies between 10 key target proteins and hesperidin were calculated. The molecular docking results revealed that AKT1, NOS3, HSP90AA1, ESR1, PPARG, SRC, ALB, MMP2, MMP9, and IGF1 exhibited binding energies of −10.6, −10.2, −9.4, −9.0, −8.8, −8.3, −7.8, −7.6, −6.7, and −6.0 kcal/mol, respectively. Generally, a binding energy lower than −5.0 kcal/mol indicates favorable binding activity between the molecule and the target protein. In this case, all the core target proteins demonstrated binding energies below −5.0 kcal/mol, suggesting a strong interaction between hesperidin and the core targets. The visualization analysis was performed using the PyMOL 2.0 software, as shown in Figure 3.

### 2.3. Molecular Dynamics Simulation

Molecular dynamics simulations are employed to validate the stability of compound–target protein binding in molecular docking. In this study, a CHARMM force field was assigned to the system, and a simulation sampling of 200 ns was performed to investigate the structural stability of the compound–target protein during the molecular dynamics simulation. The simulation trajectory was analyzed to evaluate the root-mean-square deviation (RMSD), root-mean-square fluctuation (RMSF), and protein gyration radius (RG). RMSD serves as a standard for assessing the stability of a system. During the simulation process, the RMSD value remains consistently low, indicating a close binding between the ligand and the receptor, resulting in a stable complex. Among these systems, the NOS3-hesperidin complex exhibits more variations compared to others but still falls within a stable range (Figure 4A). RMSF is employed to characterize the spatial fluctuations of small molecule ligands within the protein structure. According to Figure 4B, most receptor residues involved in the binding of naringin to the target protein remain within a stable range. However, during the simulation process in the 300–700 region, the NOS3-hesperidin complex displays significant fluctuations in RMSF values. Finally, the gyration radius is utilized to represent the compactness of the protein structure during the simulation process. Analysis from Figure 4C reveals that the protein exhibits a smaller gyration radius, indicating a tighter protein structure and a stable complex.

### 2.4. The Effect of Hesperidin on Osteoblast Proliferation

MTT was used to analyze the potentially toxic effects of different concentrations of hesperidin on MC3T3-E1 cells. In contrast to the control group, the dexamethasone-induced model group demonstrated a decline in cell viability, suggesting the successful establishment of a dexamethasone-induced MC3T3-E1 cell OP model. Compared to the control group, the concentration of hesperidin showed a concentration-dependent increase within the range of 0.3125–10 μg/mL, with a decreasing trend observed when the concentration exceeded 10 μg/mL. Furthermore, when compared to the positive drug, it exhibited favorable proliferative effects. This observation suggests that hesperidin facilitates the proliferation of MC3T3-E1 cells (Figure 5A).

### 2.5. Effect of Hesperidin on ALP Activity in MC3T3-E1 Cells

After 7 days, the experimental group treated with hesperidin exhibited a noteworthy augmentation in ALP expression, as observed through qualitative analysis using ALP staining, in comparison to the control group (see Figure 5C). Furthermore, this result was validated through quantitative analysis of ALP activity, demonstrating that hesperidin enhances ALP activity and promotes osteogenic differentiation. All experimental groups exhibited significantly elevated ALP activity compared to the control group, with higher concentrations of hesperidin correlating to greater ALP activity (*p* < 0.01), as depicted in Figure 5B.

### 2.6. The Influence of Hesperidin on the Levels of NO and Inflammatory Cytokines TNF-α and IL-6 in MC3T3-E1 Cells

Based on Figure 6, it is evident that the cellular NO levels in the model group decreased significantly in comparison to the control group. In contrast, a significant rise in TNF-α and IL-6 levels was observed. When compared to the model group, the administration of different doses of hesperidin led to an increase in cellular NO levels and a decrease in TNF-α and IL-6 levels, demonstrating a concentration-dependent correlation. Specifically, higher doses of naringin exhibited a more pronounced ability to stimulate the release of NO from MC3T3-E1 cells, as well as a more substantial inhibition of the inflammatory factors TNF-α and IL-6.

### 2.7. Toxicity Results of Zebrafish

According to Figure 6A, the toxicity results of hesperidin indicate that concentrations of 1.25, 2.5, 5, and 10 μg/mL of hesperidin do not exhibit lethal toxicity toward zebrafish larvae. The toxicity results of different concentrations of prednisolone on zebrafish larvae suggest that high concentrations of prednisolone significantly affect the survival rate of zebrafish larvae, particularly when the concentration exceeds 25 mmol/L.

### 2.8. Hesperidin has a Protective Effect on Bone Formation in Zebrafish

Following a 5-day administration, the zebrafish group that was induced with prednisolone demonstrated a notable decrease in the relative fluorescent area within the vertebral bones, in comparison to the control group. This decrease serves as evidence for the successful establishment of an OP model in zebrafish through the utilization of prednisolone. Compared to the model group, the hesperidin group showed a significantly higher relative fluorescence area, especially at concentrations exceeding 5 μg/mL. As the concentration of hesperidin increased, the fluorescence area of zebrafish approached that of the positive control group. This indicates a notable effect of hesperidin on zebrafish bone mineralization. See Figure 7B.

### 2.9. The Impact of AAPH on the Survival Rate of Zebrafish

As shown in Figure 8A, varying concentrations of AAPH in different experiments have a certain impact on embryo viability, exhibiting a concentration-dependent decrease. When induced with high concentrations of AAPH (25, 30 mmol/L), the embryo survival rate falls below 50%, making it unsuitable for further testing. Therefore, a 20 mmol/L AAPH induction solution was chosen.

### 2.10. The Impact of Hesperidin on Oxidative Stress in Zebrafish

#### 2.10.1. Effects of Varying Concentrations of Hesperidin on the Survival Rate of Zebrafish Embryos Induced by AAPH

The survival rate of zebrafish embryos in the AAPH-induced group was found to be lower compared to the control group. Hesperidin, at varying concentrations, demonstrated a notable inhibitory impact on the decreased survival rate induced by AAPH, exhibiting a response that was dependent on the concentration. The experimental results demonstrate that hesperidin provides effective protection against oxidative damage induced by AAPH in the zebrafish model (Figure 8B).

#### 2.10.2. The Effect of Different Concentrations of Hesperidin on the Production Rate of ROS Induced by AAPH in Zebrafish Larvae Was Investigated

Through the investigation of the scavenging effect of hesperidin on AAPH-induced ROS production, as shown in Figure 8C, the results indicate that hesperidin exhibits inhibitory effects on ROS in zebrafish larvae. Compared to the control group, the ROS generation in AAPH-induced zebrafish increased by 273.63%. However, treatment with hesperidin at concentrations of 1.25, 2.5, 5, and 10 μg/mL resulted in reductions of ROS generation to 236.52%, 218.21%, 182.30%, and 146.49%, respectively, demonstrating a concentration-dependent significant decrease in ROS production. Therefore, it can be concluded that hesperidin provides protective effects against AAPH-induced ROS damage in zebrafish.

### 2.11. The Impact of Hesperidin on the Expression Levels of Osteoblast-Related Proteins

Compared to the control group, the mRNA expression of ESR1, SRC, AKT1, and NOS3 in the model group cells showed a significant decrease. In contrast to the model group, the treatment group exhibited a significant upregulation of ESR1, SRC, AKT1, and NOS3 mRNA expression, with statistically significant differences observed as shown in Figure 9.

## 3. Discussion

OP is a highly prevalent disease, and its incidence is increasing annually with the growing aging population. Presently, the predominant pharmacological interventions for OP consist of chemical-based drugs, which exhibit restricted long-term effectiveness and may induce unfavorable reactions. Hesperidin, a flavonoid compound found in citrus peels, is a natural plant chemical with therapeutic potential in controlling bone metabolism disorders such as OP and osteoarthritis. This study employs network pharmacology to explore the prospective mechanisms underlying the therapeutic effects of hesperidin in OP treatment. Additionally, in vitro and in vivo functional assays were conducted, and the mRNA expression levels of crucial target genes in osteoblasts were evaluated.

The field of pharmacology, within the framework of network analysis, has gained significant attention in academic research. It presents innovative concepts and methodologies for the interdisciplinary exploration of artificial intelligence and traditional Chinese medicine. This study employs network pharmacology methodologies to forecast potential targets and anti-OP mechanisms. The results analysis indicates that hesperidin may achieve its effects by improving oxidative stress and exerting anti-osteoporotic activity through 33 potential targets. Further screening identified AKT1, NOS3, HSP90AA1, ESR1, PPARG, SRC, ALB, MMP2, MMP9, and IGF1 as the core targets. Extensive literature suggests that oxidative stress is a pathogenic factor in many chronic diseases, including OP [22,23,24,25]. Oxidative stress exerts an impact on bone density via the bone metabolism pathway, thereby contributing to the development of osteoporosis. This is achieved through the promotion of osteoclastogenesis, initiation of osteoblast apoptosis, suppression of osteoblast function, and inhibition of osteoprogenitor cell differentiation [26]. AKT1, one of the members of the AKT family, has been found to be associated with decreased skeletal muscle mass and shortened lifespan when its gene is knocked out in mice, leading to OP [27]. Estrogen receptor 1 (ESR1) forms the ESR1-Keap1-Nrf2 axis, which mediates stress response by suppressing oxidative stress and promoting osteoblasts. It also influences bone mineralization downstream of Wnt/β-catenin signaling [28,29]. Peroxisome proliferator-activated receptor gamma (PPARG) serves as a biomarker for OP, and studies have shown that inhibiting PPARG expression promotes osteogenesis in mesenchymal stem cells [30,31]. The SRC protein belongs to the SRC family kinases (SFKs) and participates in the activation of multiple signaling cascades, effectively inhibiting osteoclast formation induced by RANKL [32]. Insulin-like growth factor 1 (IGF1) and growth hormone (GH) are crucial regulatory factors in bone remodeling and metabolism, playing essential roles in acquiring and maintaining bone mass throughout life [33]. A study by Roman Garcia Pablo et al. [34] found that enhancing the GH/IGF1 axis can improve osteoblast function and slow down OP.

This present study conducted core target enrichment analysis mapping hesperidin, oxidative stress, and OP, revealing significant enrichment of the estrogen signaling pathway in the KEGG enrichment analysis. Estrogen can act on osteoblasts, osteoclasts, T cells, and bone cells to participate in bone metabolism and maintain bone formation [35]. Research has shown that the use of ERα and ERβ-specific drugs can regulate osteoblasts through estrogenic effects, thereby promoting bone formation [36]. The activation of the estrogen signaling pathway has the potential to impact the intercommunication between downstream transcription factors and the PI3K/AKT signaling pathway. The activation of the PI3K/AKT signaling pathway results in the upregulation of endothelial eNOS and the induction of elevated levels of NO, thereby stimulating angiogenesis, enhancing the local blood supply to bone tissue, and improving the microcirculatory structure. Consequently, these processes facilitate the formation of bone [37,38,39,40]. Literature reports suggest that the interaction between ER and the PI3K/AKT signaling pathway may rely on scaffold proteins and adaptor proteins binding to the p85 subunit of PI3K to form adhesive plaque protein complexes and ERα, activating downstream AKT or AKT2, thus triggering a cascade reaction in the signaling pathway [41]. Martin et al. [42] also confirmed that activated AKT can phosphorylate serine residues in the AF-1 region of ERα, thereby regulating IGF-1’s effect on Erα (See Figure 10).

In this study, we examined the impact of hesperidin on the proliferation and differentiation of MC3T3-E1 cells. The findings derived from the MTT assay demonstrate that hesperidin exhibits a concentration-dependent correlation, manifesting both proliferative and differentiating impacts on osteoblasts within a defined range of concentrations. To assess early osteogenic differentiation, an ALP activity assay was conducted on day 7 to measure alkaline phosphatase activity and perform staining. The experimental results indicated that hesperidin intervention in MC3T3-E1 cells resulted in higher ALP activity, promoting early differentiation in MC3T3-E1 cells. ELISA measurements revealed that by increasing NO levels and reducing inflammatory factors IL-6 and TNF-α, hesperidin inhibited inflammatory responses, influenced oxidative stress, and facilitated osteoblast differentiation, thereby exerting antioxidant effects to protect bone. In an in vivo zebrafish activity experiment, the results showed that hesperidin significantly enhanced vertebral mineralization compared to the control group, promoting bone formation. Furthermore, this study explored the in vivo antioxidant activity of hesperidin in a zebrafish oxidative stress model induced by AAPH, confirming its stronger oxidative stress protection ability and excellent antioxidant activity in zebrafish. RT-qPCR experiments further detected significant mRNA expression changes in core targets, including AKT1, NOS3, ESR1, PPARG, SRC, ALB, MMP2, and IGF1. The computational results from molecular dynamics simulations also supported the aforementioned experimental findings, further validating the stability between hesperidin and its related targets.

In conclusion, the network pharmacology predictions of hesperidin’s core targets and potential mechanisms in treating OP provide us with scientific evidence. The in vitro and in vivo experimental results further lay the foundation for studying the mechanisms of hesperidin in the treatment of OP.

## 4. Materials and Methods

### 4.1. Materials and Reagents

Hesperidin and icariin were purchased from Shanghai Yuan Ye Technology Co., Ltd. (batch number B20182, B21576 Shanghai, China), with a purity greater than 98%. DMEM culture medium (batch number 11965092), fetal bovine serum (batch number 12483020), and penicillin (100 U/mL) and streptomycin (100 µg/mL) (batch number 15140122) were obtained from GIBCO (New York, NY, USA). Dimethyl sulfoxide (DMSO) (batch number 472301-500ML), dexamethasone (batch number D8041), ascorbic acid (batch number A103533), β-glycerophosphate (batch number G5422-1), Calcein (batch number C302986), AAPH (batch number ajci23234), and 2,7-difluorofluorescein diacetate (DCFH-DA) (batch number D6470), prednisolone (batch number P0180), etidronate disodium (batch number B27158) were purchased from Beijing Chemical Factory Co., Ltd. (Beijing, China). The alkaline phosphatase assay kit and alkaline phosphatase chromogenic kit were obtained from Biyun Tian Biotechnology Co., Ltd. (batch number P0321S, C3206 Shanghai, China). The NO, TNF-α, and IL-6 enzyme-linked immunosorbent assay kits were purchased from Shanghai Enzyme-linked Biotechnology Co., Ltd. (batch number Shanghai, China).

### 4.2. Animals and Cells

The wild-type AB strain of zebrafish used in the experiment was provided by Nanjing Yishu LiHua Biotechnology Co., Ltd. (Nanjing, China). MC3T3-E1 is a mouse embryonic osteoblast cell line purchased from the Shanghai Cell Bank of the Chinese Academy of Sciences (Shanghai, China).

### 4.3. Network Pharmacology Analysis

#### 4.3.1. Hesperidin Target Identification

Hesperidin’s standardized SMILES number and “sdf” format file can be obtained by entering “hesperidin” into the PubChem database. (https://pubchem.ncbi.nlm.nih.gov/) (accessed on 5 May 2023). For predicting the potential target proteins associated with hesperidin, one can utilize the SwissTargetPrediction database (http://www.swisstargetprediction.ch/) (accessed on 5 May 2023), as well as the PharmMapper database (http://lilab-ecust.cn/pharmmapper/index.html), (accessed on 5 May 2023).

#### 4.3.2. Acquisition of Disease Targets

Relevant targets associated with “OP” and “oxidative stress” were predicted by querying the GeneCards (https://www.genecards.org/) (accessed on 6 May 2023), OMIM (https://www.omim.org/) (accessed on 6 May 2023), and TTD (https://db.idrblab.net/ttd/) (accessed on 6 May 2023) databases.

#### 4.3.3. Drug–Disease Interaction Target Identification and Construction of Protein–Protein Interaction Networks

Using Venny (https://bioinfogp.cnb.csic.es/tools/venny/) (accessed on 8 May 2023), the intersection between compound targets and disease targets was obtained, resulting in shared compound–disease targets.

The protein–protein interaction network was constructed using the STRING database (https://string-db.org/) (accessed on 8 May 2023) and exported as a “tsv” format file for visualization analysis in Cytoscape 3.8.0 software. Subsequently, the Analyze Network feature under the Tools menu in Cytoscape 3.8.0 software was used to analyze the compound–disease shared target network. Core targets with higher relevance were selected based on Betweenness Centrality, Closeness Centrality, and Degree values greater than the average for further investigation.

#### 4.3.4. GO and KEGG Enrichment Analysis

Based on the Metascape database (https://metascape.org/) (accessed on 9 May 2023), we performed gene ontology (GO) functional enrichment analysis and Kyoto Encyclopedia of Genes and Genomes (KEGG) pathway enrichment analysis. The relevant data was organized and visualized using the bioinformatics platform (https://bioinformatics.com.cn/) (accessed on 9 May 2023) to generate bubble plots and bar charts.

### 4.4. Molecular Docking

Download the relevant protein and ligand files in “PDB” and “SDF” formats, respectively, from the RCSB PDB database (https://www.rcsb.org/) (accessed on 10 May 2023). Locate the corresponding coordinates between the protein and the ligand. Convert the protein format to “PDBQT”. Also, convert the downloaded “SDF” format ligand for orange peel extract to “PDBQT”. Utilize AutoDock Vina 1.2.2 software for molecular docking and calculate the binding affinity between the receptor and the ligand. Visualize the results using PyMOL 2.0 software, demonstrating the binding process between the ligand and the receptor through hydrogen bonding and amino acid residues.

### 4.5. Molecular Dynamics Simulation

The initial conformation for molecular dynamics simulation was obtained using the results from molecular docking. Gromacs, a software for dynamic simulations, was employed, utilizing the Charmm36 force field and the TIP3P water model. A water box was established, and sodium ions were added to balance the system. During the simulation run, energy minimization of the entire system was performed to achieve a better molecular configuration. Subsequently, the system was equilibrated under both canonical ensemble (NVT) and isothermal–isobaric ensemble (NPT) at ambient temperature and pressure for a 100 ns molecular dynamics simulation. The simulation trajectory was further evaluated using RMSD, RMSF, and protein backbone radius of gyration.

### 4.6. Cells Culture

The MC3T3-E1 mouse embryonic osteoblast cells were cultured in DMEM supplemented with 10% fetal bovine serum and penicillin (100 U/mL) and streptomycin (100 µg/mL). The cell culture was maintained at a temperature of 37 °C in a CO_2_ incubator with a CO_2_ concentration of 5% and constant humidity. Upon reaching approximately 80% confluence, the cells were passaged following trypsin digestion. The experiments were conducted using MC3T3-E1 cells in the logarithmic growth phase.

### 4.7. MTT Analysis

Logarithmic growth phase cells were seeded in a 96-well plate at a density of 1 × 10^4^ cells per well, with 200 μL of culture medium in each well. The plate was then incubated at 37 °C in a CO_2_ incubator with 5% CO_2_ and constant humidity for 24 h until the cells adhered to the well surface. After cell adhesion, the cells were subjected to modeling treatment for 24 h, followed by drug administration at concentrations of 0.3125, 0.625, 1.25, 2.5, 5, 10, and 20 μg/mL for another 24 h of incubation. Subsequently, we added 10 μL of MTT (5 mg/mL) reagent to each well and incubated for 4 h. The supernatant was then discarded, and 150 μL of DMSO was added to dissolve the formazan crystals by shaking. The absorbance value (OD) at 450 nm was measured by a microplate reader. Cell viability (%) was calculated as follows: Cell Viability% = (OD value of drug/OD value of model group − 1) × 100%. Refer to Appendix A for the Time Schedule for the Experiment on the Anti-Osteoporotic Effects of Hesperidin.

### 4.8. The Effect of Hesperidin on ALP Secretion in MC3T3-E1 Cells

Logarithmic growth phase cells were seeded in 6-well plates at a density of 2 × 10^4^ cells per well. Osteogenic inducers, including dexamethasone (100 nmol/L), ascorbic acid (50 μmol/L), and β-glycerophosphate (10 mmol/L), were added along with different concentrations (1.25, 2.5, 5, and 10 μg/mL) of hesperidin. The induction was carried out for 7 days. Cell culture supernatants were collected and alkaline phosphatase (ALP) activity was measured using an ALP assay kit. The OD at 405 nm wavelength was determined for each well using an enzyme marker, and ALP activity was calculated for each concentration group.

### 4.9. MC3T3-E1 Cell ALP Staining

Logarithmic phase cells were seeded at a density of 2 × 10^4^ cells/well in a 6-well plate. Different concentrations (1.25, 2.5, 5, 10 μg/mL) of hesperidin, an osteogenic inducer, were added for 7 days of induction. After induction, staining was performed using the BCIP/NBT alkaline phosphatase chromogenic staining kit. The stained samples were observed and photographed under a microscope.

### 4.10. Effects of Hesperidin on the Contents of NO, TNF-α and IL-6 in MC3T3-E1 Cells

Log-phase cells were seeded at a density of 2×10^4^ cells/well in a 6-well plate and induced with osteogenic inducer hesperidin at different concentrations (1.25, 2.5, 5 μg/mL) for 7 days. Cell supernatants were collected and NO, TNF-α, and IL-6 levels were determined following the instructions provided with the ELISA assay kit.

### 4.11. Breeding and Cultivation of Zebrafish

Wild-type AB strain zebrafish were reared in a temperature-controlled incubator at 28.5 °C, with a photoperiod of 14 h of light and 10 h of darkness every 24 h. Freshly harvested brine shrimp was fed to the fish once in the morning and once in the evening, daily. The previous evening, healthy adult zebrafish of the wild-type AB strain were placed in a breeding tank with a partition in the middle, maintaining a ratio of 2 females to 1 male. The following morning at 7 o’clock, the partition in the breeding tank was removed, and an hour later, fertilized eggs were collected and transferred to 90 mm culture dishes filled with embryo water. After removing unfertilized eggs and contaminants, the dishes were placed in a biochemical incubator set at 28.5 °C for cultivation. At regular intervals, observations were made, contaminants and dead embryos were removed, and embryo water was replaced.

### 4.12. The Toxicity of Zebrafish

Randomly selected wild-type AB strain zebrafish embryos at 3 days post-fertilization (3 dpf) were used in this study. They were divided into different groups, including various concentrations of hesperidin and prednisolone. Each group was treated with different concentrations of hesperidin (0, 1.25, 2.5, 5, 10 μg/mL) or prednisolone (10, 15, 20, 25, 30 mmol/L). The toxicity on zebrafish embryos was assessed.

### 4.13. Calcein Labeling and Bone Formation

According to the size of the juvenile fish, 9 dpf zebrafish juveniles were immersed in a 2% calcein solution for 5 min, followed by three rinses with embryo water to remove unbound dye. The bone calcification within the vertebral region of the zebrafish was observed and photographed using a fluorescence microscope. The Image Pro Plus 6.0 image analysis software was utilized for quantitative analysis of the calcein-stained area and cumulative light density of the spinal column skeleton.

### 4.14. Oxidative Stress in Zebrafish Models

Randomly selected zebrafish embryos at 7–9 hpf were transferred to a 24-well plate and supplemented with embryo culture medium. Different concentrations of AAPH (10, 15, 20, 25, 30 mmol/L) were added to induce toxicity for the evaluation experiment. After completion of the exposure, the embryos were further cultured for 72 h, and the survival rate of zebrafish embryos was assessed.

Based on the toxicity evaluation results, a range of non-toxic concentrations of hesperidin was selected for antioxidant activity evaluation. Embryos at 7–8 hpf were treated with a blank control group, an AAPH-induced group, and an AAPH-induced + hesperidin group. After pre-incubation for 1 h, oxidation was induced using AAPH (20 mmol/L) induction solution. Determination of ROS production rate in zebrafish larvae at 3 dpf. Zebrafish larvae were subjected to fluorescence staining using diluted DCFH-DA (15 μg/mL) in an embryo culture medium. After incubation in darkness for 1 h, the fluorescence intensity of zebrafish larvae was observed using an in vivo fluorescence microscope.

### 4.15. RT-qPCR Analysis

The logarithmic growth phase of MC3T3-E1 cells was divided into the control group, model group, and treatment group after 24 h of processing. RNA was extracted and the purity and concentration were determined by measuring the absorbance values at 260 nm and 280 nm using Nano. The calculated amount of RNA was determined based on the measured concentration for the removal of genomic DNA reaction, reverse transcription reaction, and PCR amplification reaction. The PCR amplification primers were obtained from the NCBI database (https://www.ncbi.nlm.nih.gov/) (accessed on 16 June 2023) and selected from PrimerBank (https://pga.mgh.harvard.edu/primerbank/) (accessed on 16 June 2023) for the purpose of obtaining specific primers for the relevant genes. The specific primers required for MC3T3-E1 cells are provided in Table 1. Gene standardization was conducted using GAPDH as the internal reference, and the analysis was performed using the 2^−∆∆CT^ method. Table 1 presents the results of the RT-qPCR analysis, including the genes and their associated base sequences.

### 4.16. Statistical Analysis

Experiments should be repeated at least three times, and results should be expressed as mean ± SD. For statistical analysis of pairwise comparisons between the measurement groups, a single-factor analysis of variance using SPSS 20.0 software should be utilized. A significance level of *p* < 0.05 is deemed to indicate statistically significant distinctions.

## 5. Conclusions

In this study, a combination of network pharmacology, molecular docking, and molecular dynamics simulation techniques was employed to investigate the mechanism of action of hesperidin in anti-OP. Various in vitro and in vivo experiments, including MTT assay, alkaline phosphatase activity, ELISA, RT-qPCR, and zebrafish analysis, were conducted. Our research findings demonstrate that hesperidin, a flavonoid compound, promotes osteoblast proliferation and differentiation, inhibits inflammatory factors, reduces oxidative stress, and improves OP. These results provide a scientific basis for further exploring the clinical application of hesperidin in the treatment of OP.

## Figures and Tables

**Figure 1 molecules-28-06987-f001:**
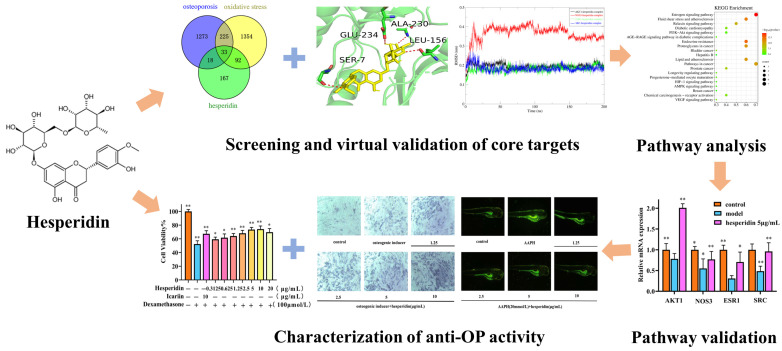
Flowchart of hesperidin anti-OP. Data is presented as means ± SD (*n* = 3). Compared to the control group, * *p* < 0.05 and ** *p* < 0.01.

**Figure 2 molecules-28-06987-f002:**
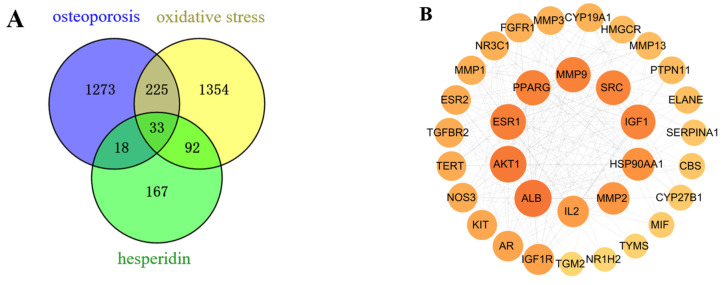
Network Pharmacology Analysis. (**A**) Venn intersection target diagram. (**B**) Protein–protein network interaction analysis. (**C**) “Hesperidin-Core Targets-Pathways-Disease” network analysis diagram. (**D**) GO enrichment analysis. (**E**) KEGG pathway enrichment analysis.

**Figure 3 molecules-28-06987-f003:**
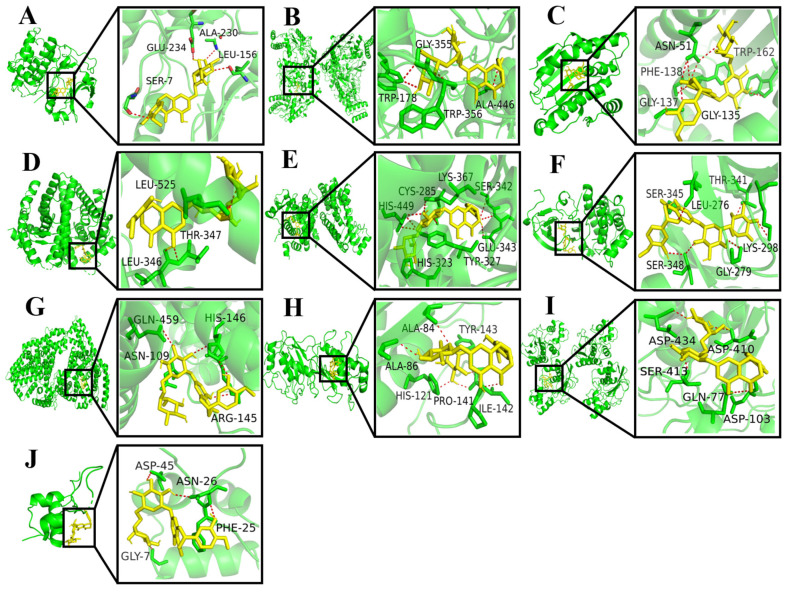
Docking results for hesperidin with core target molecules are shown. (**A**) Docking of hesperidin with AKT1. (**B**) Docking of hesperidin with NOS3. (**C**) Docking of hesperidin with HSP90AA1. (**D**) Docking of hesperidin with ESR1. (**E**) Docking of hesperidin with PPARG. (**F**) Docking of hesperidin with SRC. (**G**) Docking of hesperidin with ALB. (**H**) Docking of hesperidin with MMP2. (**I**) Docking of hesperidin with MMP9. (**J**) Docking of hesperidin with IGF1.

**Figure 4 molecules-28-06987-f004:**
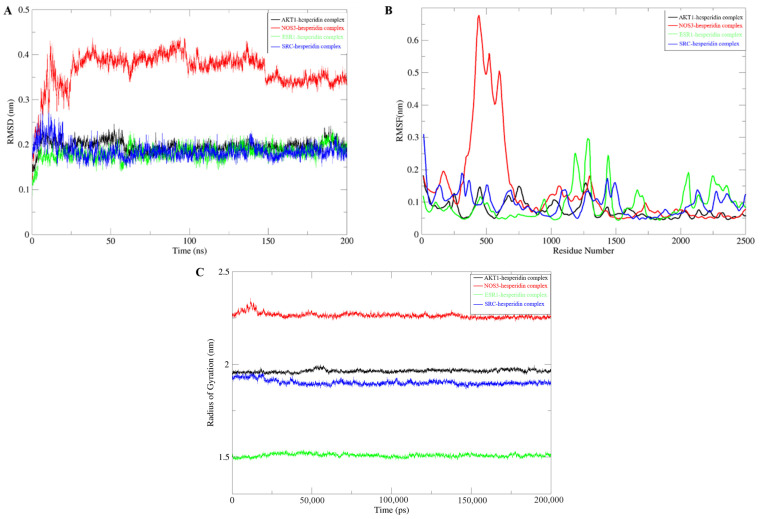
Molecular Dynamics Simulation. (**A**) RMSD Analysis. (**B**) RMSF Analysis. (**C**) Gyration Radius.

**Figure 5 molecules-28-06987-f005:**
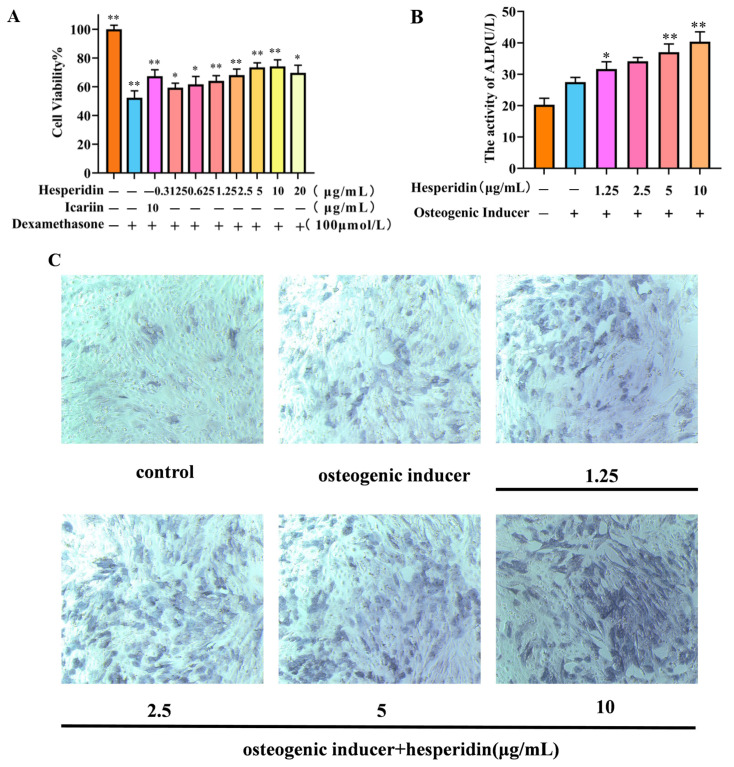
Illustrates the impact of hesperidin on MC3T3-E1 cells. (**A**) Effect of hesperidin on MC3T3-E1 cell proliferation. (**B**) Results of ALP activity. (**C**) Microscopic observation of ALP staining after 7 days. Data is presented as means ± SD (*n* = 3). Compared to the control group, * *p* < 0.05 and ** *p* < 0.01.

**Figure 6 molecules-28-06987-f006:**
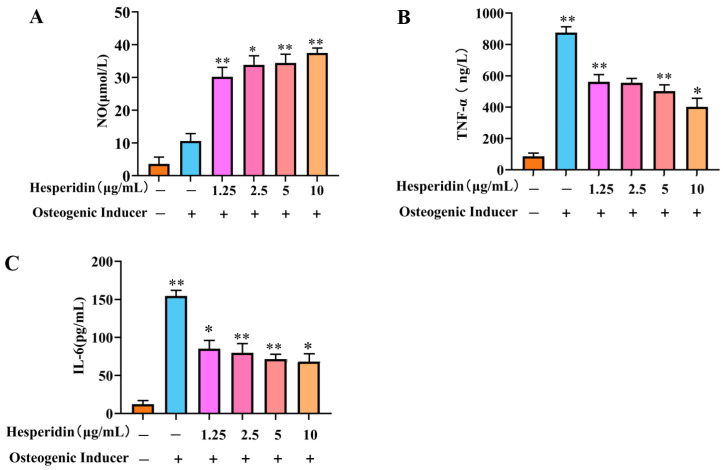
Analysis of the anti-inflammatory effect of hesperidin. (**A**) Influence of hesperidin on NO release in MC3T3-E1 cells. (**B**) Impact of hesperidin on the inflammatory factor TNF-α in MC3T3-E1 cells. (**C**) Effect of hesperidin on the inflammatory factor IL-6 in MC3T3-E1 cells. The data is presented as means ± SD (*n* = 3). Compared to the control group, * *p* < 0.05 and ** *p* < 0.01.

**Figure 7 molecules-28-06987-f007:**
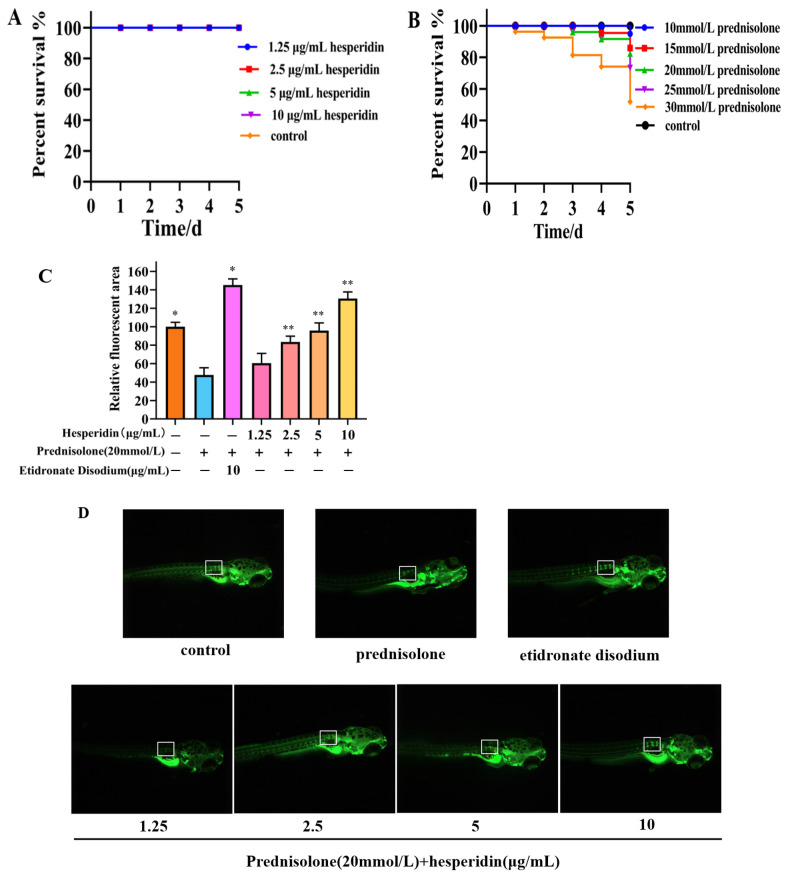
Osteoprotective effect of hesperidin in zebrafish. (**A**) Toxicity results of hesperidin in zebrafish. (**B**) The toxicity of different concentrations of prednisolone on zebrafish larvae. (**C**) The impact of hesperidin on zebrafish bone mineralization. (**D**) Calcein staining of zebrafish bones. Data are presented as means ± SD (*n* = 3). * *p* < 0.05, ** *p* < 0.01 compared to the control group. Note: The white border represents the fluorescence intensity comparison of the first three vertebral bones.

**Figure 8 molecules-28-06987-f008:**
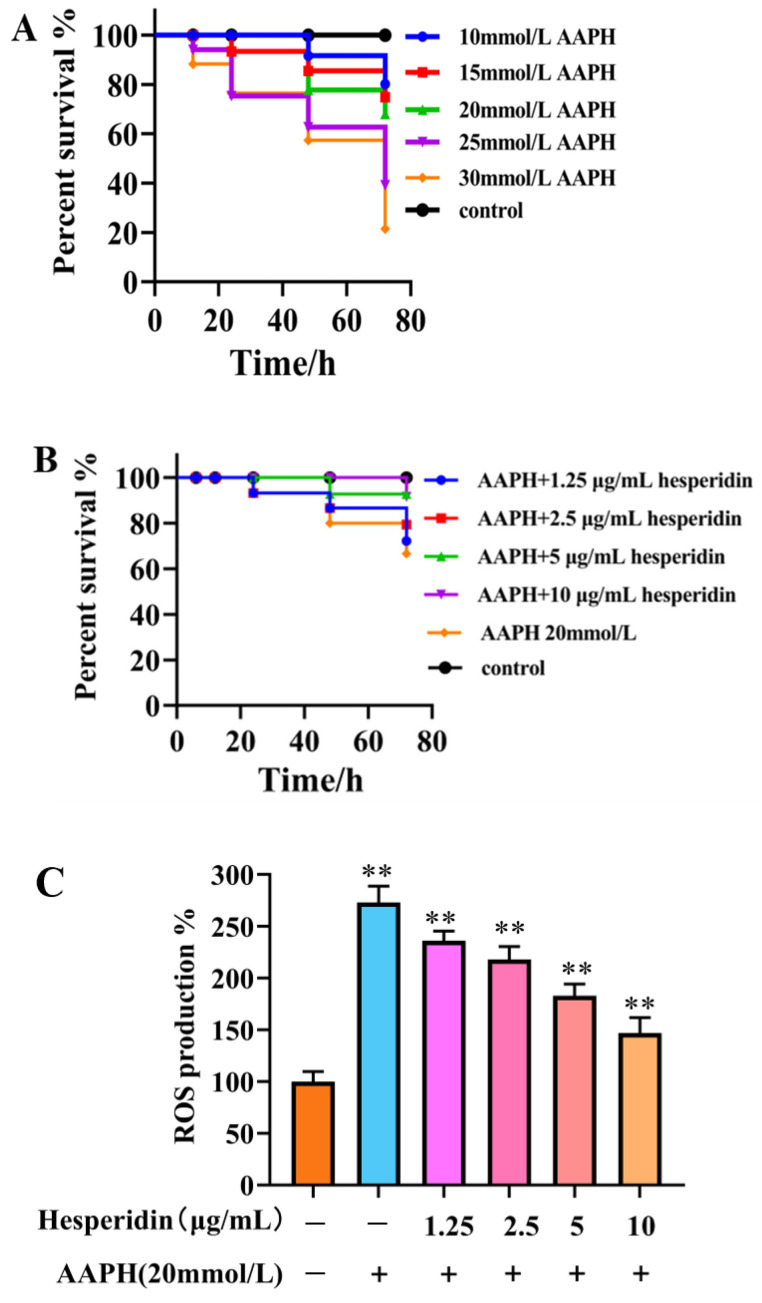
The effect of hesperidin on oxidative stress in zebrafish. (**A**) Impact of AAPH on zebrafish survival rate. (**B**) The toxicity of zebrafish larvae induced by AAPH at different concentrations of hesperidin. (**C**) The impact of different concentrations of hesperidin on the generation rate of ROS in zebrafish larvae induced by AAPH. (**D**) DCFH-DA fluorescence staining in zebrafish. Data are presented as means ± SD (*n* = 3). ** *p* < 0.01 compared to the control group.

**Figure 9 molecules-28-06987-f009:**
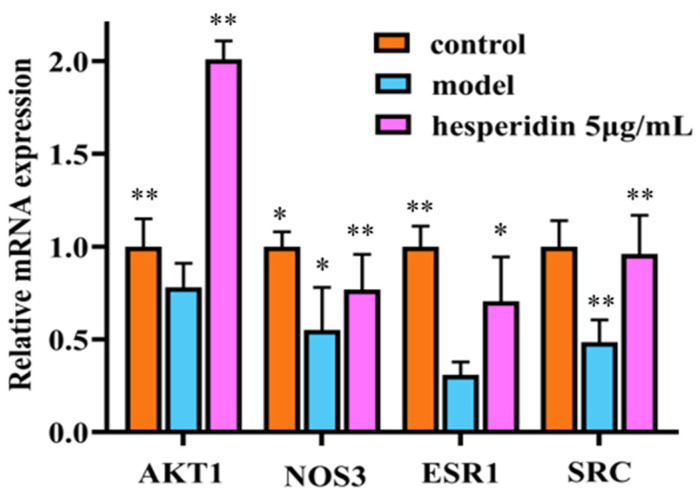
RT-qPCR analysis shows hesperidin’s impact on the mRNA expression levels of core target genes. Gene expression was normalized to GAPDH using the 2^−∆∆CT^ method. The Data were represented as the mean ± SD (*n* = 3). Compared with control, * *p* < 0.05, ** *p* < 0.01.

**Figure 10 molecules-28-06987-f010:**
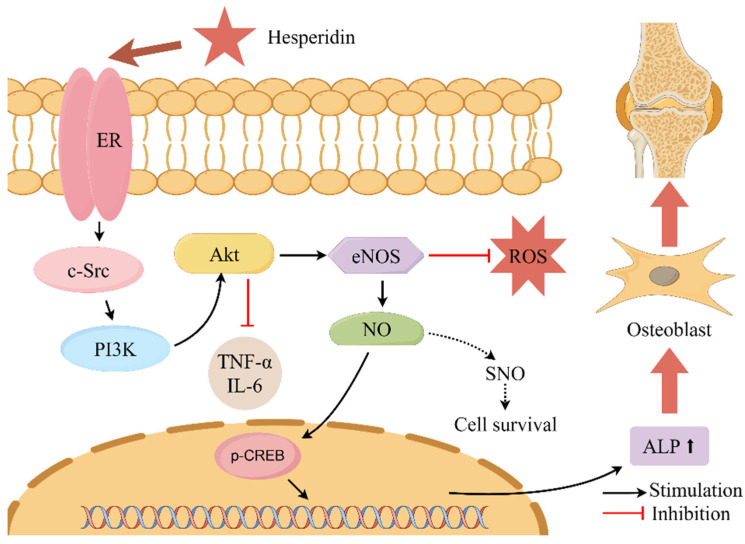
Describes the mechanism of action of hesperidin in the treatment of OP. Note: where “
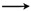
” denotes stimulation and “
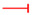
” denotes inhibition.

**Table 1 molecules-28-06987-t001:** RT-qPCR analysis of gene and associated base sequences.

Gene Name	Forward (5′-3′)	Reverse (5′-3′)
AKT1	ATGAACGACGTAGCCATTGTG	TTGTAGCCAATAAAGGTGCCAT
NOS3	TCAGCCATCACAGTGTTCCC	ATAGCCCGCATAGCGTATCAG
ESR1	TGTGTCCAGCTACAAACCAATG	CATCATGCCCACTTCGTAACA
SRC	CAATGCCAAGGGCCTAAATGT	TGTTTGGAGTAGTAAGCCACGA

## Data Availability

Not applicable.

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
