# Peer review of "Hesperidin Anti-Osteoporosis by Regulating Estrogen Signaling Pathways"

_molecules, 2023, doi:10.3390/molecules28196987_

Round 1
Reviewer 1 Report
There are various major points that need to be addressed by the authors:
-The authors applied the Hesperidin for preventing the osteoporosis and the previous findings are already showed the impact of Hesperidin on improving bone diseases etc. What the differences between the current findings and the previous reports?
-The authors used only Hesperidin without any control material for comparison, which is a major weakness of the study design.
-As the authors mentioned in the title, the verb "prevents", what are the experiments showing the prevention of the osteoporosis by Hesperidin?
-The authors are suggested to prepare a separate figure showing the time point and experimental schedule of using Hesperidin in their study.
-Figure 1 containing the findings of the paper that made it complicated and uninformative. The authors are recommended to insert details related to experimental schedule, analytical procedures etc.
-In materials and Methods section: there are various chemicals and cell lines that used in the study without mentioning the company of origin and catalog number that need to be added for repeating the experiment.
-The authors used wild-type AB strain of zebrafish as in vivo model: The authors did not mention any ethical statement or institutional approval to do the experiment that need to be provided.
- The using of zebrafish as an in vivo model... Is that finding could be transitional in human?
-What is the reason of using only MC3T3-E1 mouse embryonic osteoblast cells in the study ? Is it a strong evidence of the study finding after using this cell line of Murine origin?
-The quality of Figure 4C is too poor, and the authors are recommended to provided a figure of acceptable quality.
-The authors did not check any genes related to osteogenic differentiation and only checked ALP staining that make the conclusion of weak evidence.
-Figure 4, what the authors used as a control negative group?
-prednisolone: The authors need to provide more information of this chemical and the reason of using it.
na
Author Response
Dear reviewer,
Thank you very much for reviewing our submitted paper and providing valuable feedback and suggestions. We sincerely appreciate your thorough review and detailed comments. Based on your guidance, we have made the necessary revisions and improvements to the paper. Here is our response to the questions and suggestions you raised:
- Question: The authors applied the Hesperidin for preventing the osteoporosis and the previous findings are already showed the impact of Hesperidin on improving bone diseases etc. What the differences between the current findings and the previous reports?
Reply: Thank you for your inquiry. Our current research findings, in comparison to the previous report, primarily focus on investigating the mechanisms that promote bone formation closely associated with oxidative stress. There are several notable differences: (1) By correlating the components with disease-specific targets, we aim to elucidate the relationship between hesperidin, osteoporosis, and oxidative stress. Our main focus is to explore osteoporosis from the perspective of oxidative stress, highlighting its impact on the condition. (2) Our screening process yielded different pathways, and we specifically identified the estrogen signaling pathway. (3) In our mechanistic study, we employed MC3T3-E1 cells as the chosen cellular model to investigate their response. (4) Further in vivo experiments were conducted to validate the protective effects of hesperidin on zebrafish bone formation and oxidative stress.
- Question: The authors used only Hesperidin without any control material for comparison, which is a major weakness of the study design.
Reply: Thank you for your question. In this study, due to the lack of appropriate control materials for comparison, we supplemented the relevant positive control group in the osteoblast proliferation assay and zebrafish bone protection experiments.
- Question: As the authors mentioned in the title, the verb "prevents", what are the experiments showing the prevention of the osteoporosis by Hesperidin?
Reply: Thank you for your question. We apologize for the improper use of the term "prevention" of osteoporosis throughout the manuscript. We appreciate your valuable feedback, and in the revised submission, we have replaced it with the term "anti-osteoporosis."
- Question: The authors are suggested to prepare a separate figure showing the time point and experimental schedule of using Hesperidin in their study.
Reply: Thank you for your suggestion. We have prepared a separate figure displaying the time points and experimental timeline of the use of hesperidin in our study. It will be included as an appendix.
- Question: Figure 1 containing the findings of the paper that made it complicated and uninformative. The authors are recommended to insert details related to experimental schedule, analytical procedures etc.
Reply: Thank you for your suggestion. In order to enhance the understanding of the research process and methodology in our paper, we have revised Figure 1 and included an experimental timeline and detailed analysis procedures in the latest version of the manuscript.
- Question: In materials and Methods section: there are various chemicals and cell lines that used in the study without mentioning the company of origin and catalog number that need to be added for repeating the experiment.
Reply: Thank you for your suggestion. We have added the relevant batch numbers in the Materials and Methods section as per your recommendation.
- Question: The authors used wild-type AB strain of zebrafish as in vivo model: The authors did not mention any ethical statement or institutional approval to do the experiment that need to be provided.
Reply: Thank you for the reminder. We apologize for the omission of the ethical statement in the original manuscript. We have now included the required ethical statement in the revised version of the manuscript.
- Question: The using of zebrafish as an in vivo model... Is that finding could be transitional in human?
Reply: Thank you for your question. The zebrafish shares a high degree of homology with humans, with approximately 87% of its genes being homologous to human genes. As a species with highly similar genes to humans, the zebrafish genome and disease signaling pathways exhibit a high degree of homology to humans. Its organ development and disease physiology also show significant similarities to humans. While zebrafish serves as an intermediate model in human research, it is important to conduct further studies and validations before applying the results to humans.
- Question: What is the reason of using only MC3T3-E1 mouse embryonic osteoblast cells in the study? Is it strong evidence of the study finding after using this cell line of Murine origin?
Reply: Thank you for your question. The reason we used MC3T3-E1 mouse embryonic osteoblast cells in our study is because they are a commonly used cell line widely applied in bone biology research. The MC3T3-E1 cell line has the ability to differentiate into osteoblasts and express bone matrix-related proteins and bone-specific genes, such as alkaline phosphatase (ALP). This makes the MC3T3-E1 cell line an ideal model for studying processes such as osteoblast differentiation and bone matrix synthesis in bone cells. In addition, the MC3T3-E1 cell line is derived from mouse embryonic bone tissue, which provides good stability and reproducibility. It is well-suited for long-term culture and experimental manipulations. Finally, the MC3T3-E1 cell line has been widely utilized in bone biology research and has made significant discoveries in numerous studies. These research findings provide strong evidence and a theoretical foundation for further utilization of the MC3T3-E1 cell line.
- Question: The quality of Figure 4C is too poor, and the authors are recommended to provide a figure of acceptable quality.
Reply: Thank you for your inquiry. In Figure 4C, we have adjusted the resolution to enhance the visibility of image details, resulting in a clearer visual representation.
- Question: The authors did not check any genes related to osteogenic differentiation and only checked ALP staining that make the conclusion of weak evidence.
Reply: Thank you for your question. In our study, ESR1 is considered as one of the genes associated with osteogenic differentiation. We analyzed the expression level of ESR1 in osteoblasts using PCR. ALP, as one of the osteoblast-specific matrix proteins, is an important indicator of bone formation. It plays a crucial role in skeletal development and bone metabolism. Therefore, the activity level of ALP can be used to assess the activity of osteoblasts and the rate of bone formation.
- Question: Figure 4, what the authors used as a control negative group?
Reply: Thank you for your question. In Figure 4, we cultured osteoblasts in a complete growth medium as the control group.
- Question: prednisolone: The authors need to provide more information of this chemical and the reason of using it.
Reply: Thank you for your suggestion. We would appreciate more information regarding the chemical substance, prednisolone, and the rationale behind its usage. Prednisolone, as a glucocorticoid medication, is used as a modeling agent to establish a rapid and efficient zebrafish model of osteoporosis. Based on data mining and relevant research findings, it has been demonstrated that glucocorticoid-induced zebrafish models of osteoporosis not only lead to a reduction in robust bone mass but also result in significant changes in gene expression. In the revised manuscript, we will provide additional information in the Materials and Methods section and supplement the reasons for utilization in the introduction.
Thank you once again for your review and valuable feedback. We greatly appreciate your expertise and patient guidance. These modifications and improvements will undoubtedly enhance the quality and accuracy of the paper, which is immensely helpful for our research work. If you have any further questions or suggestions, please feel free to let us know. We would be more than happy to address them and make the necessary revisions accordingly.
Thank you once again for your time and effort!
Warmest greetings,
Wei Feng
Reviewer 2 Report
The paper is valuable and well planned.
The following changes may improve the work:
Line 430: "1% penicillin-streptomycin" should be replaced by the final concentrations of penicillin (100 U/mL) and streptomycin (100 µg/mL).
Line 441: "10µL of MTT" should be replaced by final concentration of MTT.
Line 444: The absorbance was probably measured by microplate reader.
Line 465: TNF and IL-6 levels were measured by ELISA but NO was probably measured using Griess method.
In my opinion, the paper can be published in Molecules.
Author Response
Dear reviewer,
Thank you very much for reviewing our submitted paper and providing valuable feedback and suggestions. We sincerely appreciate your thorough review and detailed comments. Based on your guidance, we have made the necessary revisions and improvements to the paper. Here is our response to the questions and suggestions you raised:
- Question: Line 430: "1% penicillin-streptomycin" should be replaced by the final concentrations of penicillin (100 U/mL) and streptomycin (100 µg/mL).
Reply: Thank you for your reminder. In our revised manuscript, it has been replaced with: penicillin (100 U/mL) and streptomycin (100 µg/mL).
- Question: Line 441: "10µL of MTT" should be replaced by final concentration of MTT.
Reply: Thank you for your suggestion. We have replaced the final concentration of MTT with 5mg/mL.
- Question: Line 444: The absorbance was probably measured by microplate reader.
Reply: Thank you for your reminder. The absorbance was measured using a microplate reader. I appreciate your input, and in our revised manuscript, we have made the necessary substitution.
- Question: Line 465: TNF and IL-6 levels were measured by ELISA but NO was probably measured using Griess method.
Reply: Thank you for your reminder. Indeed, in our current measurement process, NO was also measured using an ELISA assay kit. I appreciate your input, and we have made the necessary adjustment in our revised manuscript.
Thank you once again for your review and valuable feedback. We greatly appreciate your expertise and patient guidance. These modifications and improvements will undoubtedly enhance the quality and accuracy of our paper. Your assistance has been invaluable to our research work. If you have any further questions or suggestions, please don't hesitate to let us know. We would be more than happy to address them and make the necessary revisions.
Thank you once again for your time and effort!
Warmest greetings,
Wei Feng
Reviewer 3 Report
It's an interesting job, describing with bioinformatics approaches the possible mechanisms associated with the effect of Hesperidin, which are subsequently corroborated in vitro.
My only observations are related to justifying why you evaluated AKT1, NOS3, ESR1, SRC markers, out of the 10 that you identified in silico. What was your criteria for selection?
Also, please mention how many times the molecular docking was performed to obtain the binding energy values. Do the results show an average of these analyses?
Finally, regarding the order, I believe the information would flow better if you placed the results of the molecular dynamics immediately after the docking.
Author Response
Dear reviewer,
Thank you very much for reviewing our submitted paper and providing valuable feedback and suggestions. We sincerely appreciate your thorough review and detailed comments. Based on your guidance, we have made the necessary revisions and improvements to the paper. Here is our response to the questions and suggestions you raised:
- Question: It's an interesting job, describing with bioinformatics approaches the possible mechanisms associated with the effect of Hesperidin, which are subsequently corroborated in vitro.
Reply: Thank you for your affirmation. We sincerely appreciate your feedback and suggestions.
- Question: My only observations are related to justifying why you evaluated AKT1, NOS3, ESR1, SRC markers, out of the 10 that you identified in silico. What was your criteria for selection?
Reply: Thank you for your inquiry. The criteria for selecting the AKT1, NOS3, ESR1, and SRC targets among the 10 core targets are based on network pharmacology analysis. It was found that these targets ranked highly in protein-protein interaction (PPI) analysis. Additionally, it was observed that AKT1, NOS3, ESR1, and SRC proteins are concentrated and enriched in a specific pathway. Therefore, we focused our research on these four targets and conducted further validation.
- Question: Also, please mention how many times the molecular docking was performed to obtain the binding energy values. Do the results show an average of these analyses?
Reply: In the molecular docking section, the binding energies of hesperidin compounds with 10 proteins were analyzed by performing molecular docking simulations using AutoDock Vina software. The simulations were run 200 times, and from each run, the top 9 conformations with the lowest binding energies were obtained. The reported results represent the binding energy values of the best conformations among the 9 obtained conformations.
- Question: Finally, regarding the order, I believe the information would flow better if you placed the results of the molecular dynamics immediately after the docking.
Reply: Thank you for your suggestion. We have reorganized the latest submitted manuscript by placing the section on molecular docking after the results of molecular docking, thus providing supplementary information to the molecular docking results.
Thank you once again for your editorial work and invaluable feedback. We deeply appreciate your professional expertise and patient guidance. These revisions and improvements will undoubtedly enhance the quality and accuracy of the paper, greatly benefiting our research work. If you have any further inquiries or suggestions, please do not hesitate to inform us. We would be delighted to address them and make the necessary modifications accordingly.
Thank you once again for your time and effort!
Warmest greetings,
Wei Feng
Reviewer 4 Report
The authors are investigating the effects of the flavonoid hesperidin on osteogenesis and oxidative stress. Specifically, they used network pharmacology to predict the core targets and signalling pathways involved in anti-osteoporosis effects and examined their effects on bone formation using osteoblasts MC3T3-E1 and zebrafish. The results showed that hesperidin alleviates osteoporosis by activating estrogen signaling pathways via ESR1, upregulating SRC, AKT, and eNOS, inducing increased NO levels, promoting osteogenesis, and suppressing inflammatory cytokines to alleviate oxidative stress, thereby the authors conclude that the study has shown the potential to exert an effect to alleviate osteoporosis. The work is carefully carried out and deserves to be published essentially as it stands. It should be possible to accept this paper once the following points have been more fully addressed.
1) The flowchart (Fig. 1.) is considered unnecessary because it is redundant with the text. Please delete it. However, the structural formula of hesperidin should be included in the paper.
2) Fig. 6. D: The orientation of the zebrafish photos is not consistent. We also think that fluorescence by calcein is attenuated in the prednisolone-treated group compared to the control group, but this does not appear to be the case in the pictures in Fig. 6.
3) Fig. 7 B: The concentration of AAPH should be described.
4) Fig. 8: Please indicate in the graph the concentration of hesperidin used in the RT-qPCR experiment.
Author Response
Dear reviewer,
Thank you very much for reviewing our submitted paper and providing valuable feedback and suggestions. We sincerely appreciate your thorough review and detailed comments. Based on your guidance, we have made the necessary revisions and improvements to the paper. Here is our response to the questions and suggestions you raised:
- Question: The flowchart (Fig. 1.) is considered unnecessary because it is redundant with the text. Please delete it. However, the structural formula of hesperidin should be included in the paper.1)
Reply: Thank you for your suggestion. We have removed Figure 1 in the revised manuscript and retained the structural formula of hesperidin.
- Question: Fig. 6. D: The orientation of the zebrafish photos is not consistent. We also think that fluorescence by calcein is attenuated in the prednisolone-treated group compared to the control group, but this does not appear to be the case in the pictures in Fig. 6.2)
Reply: Thank you for bringing this to our attention. We have noticed the inconsistency in the orientation of the zebrafish images in Figure 6D. We have made the necessary modifications to Figure 6D in the revised manuscript that has been uploaded. We stained the zebrafish skeletal tissue with calcein green, which allows us to visualize mineralized zebrafish bone through green fluorescence. We observed the fluorescence intensity of vertebral bones. In Figure 6D, we have marked the location of the observed vertebral bones and found that the fluorescence intensity of the vertebral bones in the prednisolone-treated group is weaker compared to the control group.
- Question: Fig. 7 B: The concentration of AAPH should be described.3
Reply: Thank you for bringing it to our attention. We have addressed the issue regarding the description of AAPH concentration in Figure 7B and made the necessary modifications in the resubmitted manuscript.
- Question: Fig. 8: Please indicate in the graph the concentration of hesperidin used in the RT-qPCR experiment.
Reply: We appreciate your reminder, and we have taken note of the missing concentration of hesperidin in RT-qPCR in Figure 8. We have made the necessary modifications to address this in the revised version.
Thank you once again for your diligent review and invaluable feedback. We greatly appreciate your expertise and patient guidance, as these modifications and improvements will undoubtedly enhance the quality and accuracy of our paper. Your assistance is immensely valuable to our research endeavor. If you have any further questions or suggestions, please don't hesitate to let us know. We would be more than happy to address them and make the necessary adjustments accordingly.
Thank you once again for your time and effort!
Warmest greetings,
Wei Feng